# Informal workers' perceptions of retirement planning in developing countries

**Navneel Shalendra Prasad**[1]*, **Amit Prakash**[2], **Nikeel Nishkar Kumar**[2,3]

**1** Department of Management, School of Business and Economics, The University of Fiji, Lautoka, Fiji,
**2** Department of Economics, School of Business and Economics, The University of Fiji, Lautoka, Fiji,
**3** Department of Finance, School of Economics, Finance, and Marketing, RMIT University, Melbourne, Australia

☻ These authors contributed equally to this work.
\* navneelp@unifiji.ac.fj

## Abstract

In this study, we examine the perceptions of informal workers toward retirement planning in developing countries. Specifically, we examine vegetable market vendors' perceptions of the local superannuation fund, namely the Fiji National Provident Fund (FNPF) in the Western division of Fiji, where agriculture is predominant. The research aims to address the gap in the literature concerning nest egg savings options for the part of the population for whom superannuation contributions are not mandatory. Nest eggs savings are savings and investments for huge financial goals such as retirement. Generally, respondents had a positive outlook on the FNPF, and most responded that they would prefer a tailor-made retirement plan suited to the income and livelihood of market vendors. Specifically, as self-employed people, market vendors are keen for a specialized system of voluntary contributions towards their FNPF accounts that bypasses the mandatory joint-contributions from employers. We also find that age, gender, duration as a market vendor, and other savings avenues are vital for accessibility to financial services offered by the FNPF. We also find that while market vendors are aware of saving and saving benefits with FNPF, there is a lack of intention for them to adopt the current FNPF product. Policymakers must, therefore, develop FNPF products to cater for the needs of market vendors recognizing self-employment and the voluntary nature of contributions.

## Introduction

Agricultural sector development is one of the most powerful tools to end extreme poverty, boost shared prosperity, and feed around 10 billion people by 2050. Compared to other sectors, its growth is two to four times more effective in increasing the incomes of the poorest. Agriculture accounts for 4% of the global gross domestic product and more than 25% in the least developing countries [1]. In Fiji, agriculture is largely an informal sector [2]. The most recent statistics by the Fiji Bureau of Statistics indicate that the informal sector in Fiji consistently contributed as much as 23% of Fiji's GDP in 2011 [2]. The informal sector in Fiji is dominated by agricultural sector workers such as market vendors [2]. However, such workers do not contribute to superannuation schemes such as the Fiji National Provident Fund

**Data availability statement:** All data are in the paper and supporting information files.

**Funding:** This research was supported by a grant from the Fiji Higher Education Commission (FHEC) through The University of Fiji. The FHEC ensures quality assurance in Higher Education in Fiji. The University of Fiji approved this funding through the Research Committee on 18 August 2016 (Grant Number: 2016GOVGRT001). The funders had no role in study design, data collection and analysis, decision to publish, or preparation of the manuscript.

**Competing interests:** The authors have declared that no competing interests exist.

(FNPF), have no formal contract letters guaranteeing employment, and their employment is often not in compliance with Fijian labour laws [2]. Retirement planning for informal workers such as market vendors is therefore imperative.

A critical first step in assessing the retirement planning of market vendors is ensuring sufficient funds for old age. Market vendors as informal workers are excluded from social safety nets and Government protection schemes [2]. Informal workers in Fiji are excluded from retirement benefits unless they voluntarily contribute towards their FNPF accounts [2]. Superannuation programs such as the FNPF are available to informal workers such as market vendors on a voluntary basis, and not as a mandate. Based on our survey, the majority of the respondents did not save in FNPF despite being aware of FNPF and its purported benefits. This speaks to a plausible lack of financial literacy motivating this study.

A superannuation scheme is a fund where workers regularly contribute towards a future pension and retirement nest egg. Workers from formal sectors have access to superannuation where they and their employers make mandatory contributions. In brief, the private superannuation landscape in Fiji is described as follows. Mandatory members contribute 8% of their income towards their FNPF, with an additional contribution of 10% from employers. During COVID-19, the government reduced these contributions to 5% respectively between 1 April 2020 and 31 December 2021. These contributions were raised to 6% each in 2022 and 7% in 2023. The 10% and 8% contributions were normalised from 1 January 2024 [3].

The savings benefits with FNPF include retirement and pre-retirement benefits such as housing, medical, education, funeral, and unemployment assistance. The fund allocates members' contributions to the Preserved Account and General Account. The Preserved Account accumulates 70% of all contributions, while the remaining 30% is accounted for in the General Account. All the pre-retirement benefits are distributed from the General Account except for housing. On a member's purchase of their first property, they are entitled to all the contributions in the general account and 30% from the Preserved Account. In 2022, there were a total of 398,593 members in FNPF, 214,536 of whom were compulsory members, 12,242 were voluntary members, and the remainder were dormant members. Of all voluntary members, 55% were females, and 45% were males. There were 6,632 voluntary members in 2021 compared to 12,242 in 2022. Voluntary members contributed 4.8 million in 2021 and 20.1 million in 2022 [4].

Those working in the informal sector, self-employed and working overseas can contribute to the fund as voluntary members with a $10 opening deposit and a minimum of $10 deposit per transaction. Payments in the voluntary membership can be made through bank transfers, payments at FNPF outlets, payments at Post Fiji (postal service providers with offices in diverse locations) outlets and mobile payments. Employers of the informal workers, if any, are not required to pay the mandated employer contributions as with the formal sector [5]. Our survey implies that market vendors generally appear to lack technical knowledge on how voluntary contributions can be made despite having an awareness of FNPF and its benefits.

This study examines the perception of market vendors towards superannuation programs in Fiji. Retirement planning is a crucial part of financial planning to sustain the livelihood of the workers engaged in the agriculture sector. A sound retirement plan such as clearly defined FNPF policies for market vendors may encourage participation in the largely informal agriculture sector. There may be many other methods to foster incentives via government-assisted programs. The Fiji government provides six main social assistance programs. These include the poverty benefit scheme, care and protection allowance, social pension scheme, disability allowance, rural pregnant mother's food vouchers, and the bus fare scheme [6]. However, these are macroeconomic efforts with little to no effort directed at the welfare effects of the agriculture sector participants at the grassroots level. Notably, the informal economy, such as

the market vendors, is characterized by hand-to-mouth behaviour with little financial discipline to save [7].

The Reserve Bank of Fiji states a need for financial inclusion policies by the government to incentivize financial products and models aimed at financially excluded populations [8], such as farmers who do not have access to superannuation. This may deter people from considering farming as a long-term career because the presence of mandatory employer and employee contributions on top of baseline salary makes formal employment lucrative. There is a need for formal FNPF initiatives in the informal sector because Fiji is a developing country and like other similar countries, is plagued with issues of poverty, rising inequality, and imports of key goods and services. As a starting point, addressing disincentives within the informal sector may encourage people to enter these industries which may have beneficial flow-through economic effects on local agricultural production helping curb imports [2].

Accordingly, in this study, we provide survey-level evidence on the perceptions of market vendors in the western division of Fiji towards the national superannuation fund, Fiji National Provident Fund (FNPF). The scope of the paper is that we look at voluntary pensions and the plausible reliance of market vendors on such schemes for their retirement. The survey design is intended to answer our primary research questions: Is there a savings plan for small market vendors? Do these small market vendors have a mapped retirement plan for old age? Consequently, the two main objectives of this study are (1) what are the determinants of the awareness of FNPF and its benefits by market vendors, and (2) are there tailor-made FNPF programs catering for market vendors in Fiji? The first objective is fulfilled by regression analysis. The second objective is fulfilled by interviews and thematic analysis.

The study contributes to research on superannuation schemes focusing on the informal sector in developing countries. To the best of our knowledge, prior research has not examined the determinants of FNPF awareness or its benefits in informal sector workers within the specific context of Fiji. Market vendors as informal sector workers are underrepresented in research and outreach by the FNPF. By conducting a study in this setting, we gain primary first-hand information on the ideas and views of market vendors towards superannuation. Financial planning and, specifically, retirement planning in agriculture is important because it is a crucial step that brings economies closer to achieving the United Nations' sustainable development goals, reducing poverty, providing decent work, and reducing inequalities for informal workers such as farmers. Apart from scholarly interest, the study provides important insights that can help curb social problems by providing plausible solutions for disincentives in agriculture, retirement planning and financial security for the informal sector in Fiji. In this sense, addressing the critical financial disincentive in the agriculture sector for individual farmers may curb the decline in agricultural participation [9].

## Literature review

### Informal sector, superannuation and financial literacy

The informal sector contributes to providing essential products and services and generating employment [10]. In developing economies, the informal sector increased its size from 37% of the Gross Domestic Product (GDP) in the 1990s to approximately 50% by 2010 [11]. Informal employment accounts for more than half of all jobs in Fiji, Palau, Tonga, and Vanuatu [12]. The review by [13] suggests that research in the informal sector remains underexplored. The authors argue that management scholarship has enormous potential to improve understanding of the informal sector. Van Ginnekan [14] notes that informal workers contribute to a larger and increasing part of the labour force in developing countries. Many of these workers are unable or unwilling to contribute a percentage of their income to utilize formal sector

social benefits. Thus, these workers set up their own health and social benefits schemes aligning with their contribution capacity. Special social assistance schemes are required to protect the most vulnerable in this group.

Hu and Stewart [15] provide an overview of pension coverage for informal sector workers. Governments use two approaches, means-tested and universal, to provide social assistance to the poorest elderly on a non-contributory basis. In the means-tested approach, only people who are too poor to support themselves are covered, while the universal approach covers all elderly populations, regardless of income level. South Africa and Bangladesh use the means-tested approach. In South Africa, this pension is the primary source of income for 75% of retirees. In China, it is mandatory for informal sector workers to join a pension scheme. However, very few comply with this requirement due to a lack of incentives, high contributions and low enforcement powers of the government. In many local regions in China, restrictions have been eased to encourage participation. There are many voluntary pension schemes in China. It is noted that simplification and flexibility should be the main features of new pension products. In Chile and other countries with a more significant agriculture sector, flexible contributions are allowed. Flexibility of withdrawals should also be a feature in new pension products as the informal sector workers are a vulnerable group. Lee et al. [16] find that the informal sector in Mexico makes retirement decisions mainly based on health and access to health insurance through social security and not based on socioeconomic benefits. Onyango et al. [17] found that informal sector workers in Nairobi, Kenya, recognized the benefits of savings. However, they focused on meeting the needs of their immediate lives and did not consider the future, let alone retirement, as old age was perceived as bleak, dull, restricted and not worth considering. Tran and Jung [18] find that the effect of public pensions is two and a half times larger when accounting for the interaction between private transfers and the elderly's retirement decisions.

Financial literacy relates to retirement planning and superannuation. Financial literacy refers to the effective use of various financial planning skills to budget and plan personal finances, as well as being aware of how financial products such as savings accounts work [19]. People's knowledge of and ability to use fundamental financial concepts in their economic decision-making matters as this often has longer-term repercussions on their lives and livelihood [19]. Having knowledge of financial products is, however, only part of the picture. True financial literacy may relate to having a working knowledge of the mechanics of financial products, such as understanding rates of return and how this relates to inflation rates and general interest rates in the economy [19]. A lack of financial literacy, therefore, underscores that market vendors may not have FNPF accounts because they may not understand how such accounts function.

A lack of financial literacy undermines financial inclusion, especially in developing countries. Financial inclusion ensures that individuals, especially poor people, access essential financial services in the formal financial sector [20,21]. Ozili [22] finds that financial inclusion is influenced by the level of financial innovation, poverty levels, the financial sector's stability, the state of the economy, financial literacy, and regulatory frameworks, which differ across countries. Financial inclusion has received academic attention for four reasons. First, it is considered a major strategy to achieve the United Nations' sustainable development goals [23]. Second, it helps to improve social inclusion [24]. Third, it can reduce poverty [25]. Fourth, it brings socio-economic benefits [26]. The government used financial inclusion in Argentina to draw people into the formal banking system. People began to use more debit and credit cards, which led to more consumption in the formal markets, which the government can easily tax [27]. In Bangladesh, financial inclusion was achieved through financial innovations such as SureCash (Mobile Money Transfer), which reached women and the poor [28]. In developing

Asia, less than 27% of people have an account in a formal financial institution, and only 33% of enterprises have credit or loans from a financial institution. High costs, geographic access and lack of identification were major barriers to financial inclusion in developing Asia [29]. In Australia, remote Indigenous groups were the most financially and digitally excluded. While many of these communities have mobile phones (half of which are smartphones), mobile phone banking is unpopular [30]. The link between these groupings is that informal sector workers may lack financial literacy. As such, they may have comparatively less knowledge of financial planning, such as superannuation schemes for retirement planning. This is expected to be a severe problem in developing countries where the informal sector is more prevalent. Discussing superannuation schemes in other developing countries helps situate our paper within the broader literature focusing on superannuation schemes in developing countries.

Existing investment opportunities and support for farmers in Fiji come through the Department of Cooperatives. The department, introduced in 1947, focuses on promoting the development and establishment of rural Agri-based production [31]. Sugar cane farmers form cooperatives and buy sugarcane harvesters in many communities across, something impossible to achieve by an individual or small group of farmers [32,33]. Cooperatives alone will be unable to solve Fiji farmers' retirement issues. Another option for farmers in Fiji to invest is trust funds. Fiji has two trust funds, the Unit Trust of Fiji [34] and the Fijian Holdings Unit Trust [35]. Investing in these trust funds allows you to earn dividends and growth of the price of your units over time. In this study, we enquire about other avenues of savings that market vendors use other than FNPF.

## Retirement planning and superannuation in developing countries

Superannuation is a form of retirement planning which can be classified as long-term financial planning. A superannuation is the compulsory deduction of a specified amount of labour compensation to accumulate a retirement nest egg. This deduction exists only for those who hold jobs at either private or public institutions. Jefferson and Preston [36] argue that increasing personal savings is an alternative to superannuation that can be considered for those who do not know of superannuation contributions. This is a plausible alternative for those individuals who are informal workers, such as farmers. Although not without drawbacks, individuals may need to withdraw these accumulated savings, which is counterproductive. Altfest [37] presents an overview of the growing field of personal financial planning (PFP) and attributes its origins to prominent economists such as Modigliani, Becker, and Markowitz.

PFP comprises all items of financial interest to an individual. These include tax planning, cash flow planning, investments, risk management, retirement planning and estate planning [37]. Many individuals do not utilize these financial instruments, and a leading explanation for this is that people are not financially literate [38]. This can be extended to market vendors in Fiji. Due to the lack of financial instruments and financial education, market vendors are generally ignored in Fiji regarding superannuation schemes. Thus, we provide first-hand evidence of PFP in Pacific Island countries lacking financial institutions to support the marginalised vendor community. We study the awareness level of market vendors in Fiji of superannuation as a means of saving and the benefits of saving in a superannuation fund.

Retirement planning broadly relates to financial literacy. There are numerous literatures on well-established financial literacy programs. However, most case studies are unique to the USA. The scarcity of basic financial concepts harms retirement planning, participation in stock markets, and borrowing behaviour [39]. Lusardi and Mitchell [40] used their 2004 survey by including the US Health and Retirement Study (HRS) to investigate the rationale for workers' decision toward retirement savings avenues for collecting relevant financial

literacy information and their level of financial literacy. They concluded that older Americans (50 years or older) were more likely to be financially illiterate, which hindered their financial knowledge, a crucial element for financial planning.

Furthermore, individuals are more comfortable using formal methods of collecting financial information like retirement calculators, retirement seminars, and financial experts rather than family, relatives, or co-workers. Similarly, Lusardi and Mitchell [41] used the 2004 HRS study and discovered a strong correlation between financial planning, financial literacy, and housing wealth. They also found that financial planning strongly correlates with financial and political literacy. Following this, Lusardi and Mitchell [42] investigated financial planning using the new Rand American Life Panel (ALP). They found a consistent result with previous evidence from HRS (positive correlation between knowledge of financial matters and retirement planning). Bucher-Koenen and Lusardi [43] explored the relationship between financial literacy and retirement planning in Germany and found that financial literacy had a significant influence on an individual's retirement planning. Similarly, the importance of financial literacy in retirement planning in the Netherlands using household survey data was justified by Van-Rooij et al. [44]. In the latter research, Van-Rooij et al. [45] used information on smoking and heavy alcohol drinking as a proxy for myopic behaviour, and the results were like previous research. In addition, the studies by Klapper et al. [46] in Russia and Fornero and Monticone [47] in Italy supported the casualty from financial literacy to retirement planning.

Agarwal et al. [38] contributed to financial literacy and financial planning findings by examining the investment behaviour, liability choice, risk tolerance, and insurance usage of program participants in India. The authors found that most respondents were financially literate on interest rates (numeracy), inflation, and risk/diversification but varied across demographic and socioeconomic groups. In a more recent study, Ricci and Caratelli [48] aimed to contribute new insights into Italy's case by evaluating several indicators of retirement planning and finding that trust positively influences people's decision to enter private pension schemes and to devote severance pay to a private pension scheme. Kerry [49] reviews that conceptually, retirement planning continues to be poorly delineated and, therefore, narrowly investigated. Empirically, cognitive antecedents of retirement planning remain prominent in workplace and retirement researchers.

As we discuss financial inclusion in the informal sector and lack of financial planning based on lack of financial literacy, it is also important to review superannuation adoption in other developing countries such as Fiji. In Thailand, Jansuwan and Zander [50] recommend that a pension higher than the available old-age allowance can support farmers in maintaining a better living standard after retiring. The issue, however, is that only a fraction of farmers currently have access to a pension. Older farmers in Thailand may adopt labour-saving farm technologies, and some seek less intense off-farm work. Older farmers also reduce the cultivated land by leasing or selling [51].

In Indonesia, there is meagre participation of workers in pension fund programs due to inadequate employee pension schemes or occupational pension schemes (OPS), dues that burden employees' finances [52], and a low level of knowledge about pension schemes [53]. Previous studies indicate that OPS has a positive effect on retirement intentions (RI) [54,55], OPS prevent workers from quitting [56], and the effect of OPS on RI is determined by the structure of the retirement plan [52]. Wang and Zhang [57] studied the impact of the New Rural Society Pension Insurance pensions of China on the mental well-being of the rural elderly and found that pension income improves mental well-being by relieving depression of the rural elderly however, the beneficial effects of pension income are minimal. In China, Eggleston et al. [58] indicate that the Chinese New Rural Pension Scheme (NRPS) income increases the migration of adult children but does not affect elderly living arrangements. Cheng et al. [59] conclude

that NRPS leads to adult children living nearby, higher socioeconomic status, and better health status. Liu and Zhuang [60] and Li and Sicular [61] found that pension programs not only provide the older population with additional income but can also improve land use efficiency by shifting the operation of farms away from senior farmers who have been found to be less productive. It also allows them to substitute leisure for work and reduce their dependency on agricultural production. In Sri Lanka, Heenkenda's [62] study shows that most individuals lack awareness of their Voluntary Pension Scheme (VPS) even if they are active members. The study also indicates that households with many dependents contributed more to dropouts. The study highlights that income, assets, financial inclusion, financial literacy, and social capital significantly influence individuals discontinuing their pension scheme.

This study complements the literature on retirement planning in the informal sector. While awareness of superannuation funds has not been studied previously, our study shows good awareness of superannuation and superannuation benefits by Fijian market vendors. A distinction that has not been previously made but is vital in studying intentions to use superannuation was studied in this paper. Here, the distinction is made between being aware that there is a superannuation and knowing the benefits that the superannuation offers (we can call this financial literacy). This study suggests that awareness levels do not necessarily lead to the adoption of superannuation. This is mainly because existing superannuation plans are not simple and flexible enough for market vendors and informal workers in particular. Therefore, there is a need for a tailor-made scheme for market vendors and informal workers.

## Data and methods

### Data

The methodology collects quantitative and qualitative data from vegetable market vendors in the western division of Fiji via face-to-face interviews. Based on these responses received, commonalities are identified from the responses gathered to derive conclusions about the perceptions of market vendors towards the FNPF savings scheme. Given the difficulties in gathering enough respondents for analysis, the study used a convenience sampling technique, where participants were recruited based on whoever was willing to participate [63].

Convenience sampling was used for data collection as market vendors fall under the administration of local municipal councils whose approval needs to be sought before data collection in their respective markets. Nevertheless, there are drawbacks of convenience sampling such as a potential lack of generalizability or a selection bias where the findings may apply only to the specific group sampled leading to an over-representation of certain groups as well as limiting of the scope of the survey. Nevertheless, convenience sampling is used due to difficulties in obtaining suitable data for analysis such as in developing countries like Fiji.

This was the case in our research, where approval was not always granted by municipal councils to interview market vendors. As such, we were only given permission to interview three markets in the western division of Fiji and were advised by the market administrators to approach market vendors for interviews only if they were willing to participate. We were also advised by market administrators not to ask questions that could hurt the sentiments of the market vendors, such as their level of education and sexual orientation. Even for markets where we received permission to interview market vendors, market vendors themselves were reluctant to be part of the interviews. For this reason, convenience sampling was adopted.

An ethics approval was sought from the University of Fiji Research Committee. Verbal consent was sought from market vendors before interviewing them. Nevertheless, we gave our best efforts to ensure a reasonable balance between genders within our sample. 8.4% of market vendors were between the ages of 18 and 30, about 27% between 31 and 40, about 27%

between 41 and 50, about 28% between 51 and 60, about 6% between 61 and 70 and 4% above 71. In this study, 31% of the respondents were males, while 69% were females. A major reason for this is that the majority of the market vendors are Females. Vegetable market vendors in this context are full-time market vendors owning a stall at municipal vegetable markets in the western division. The western division in this case study refers to Sigatoka, Rakiraki, and Tavua. Participants were recruited and interviewed between 2017 and 2019. Participants were informed about the research and how data will be utilised. All participants voluntarily participated, and their privacy is fully protected. The authors did not have access to any information that could identify participants during and after data collection.

## Objective 1

Objective 1 seeks to identify the determinants of the awareness of FNPF and its benefits. It is fulfilled by regression analysis. Our regression models are specified as follows:

$$
\begin{aligned}
OUTCOME = &\alpha_0 + \alpha_1 AGE + \alpha_2 LTN + \alpha_3 GNDR + \alpha_4 SIZE + \alpha_5 INC + \alpha_6 DRTN \\
&+ \alpha_7 SAVNG + \alpha_8 CURRENT\ SAVNG + \alpha_9 AVNE + \alpha_{10} FRMVND \\
&+ \alpha_{10} WYRMV + u_i
\end{aligned}
\tag{1}
$$

where OUTCOME is the dependent variable, which is either FNPF AWARENESS or FNPF BENEFIT, which measures whether market vendors are aware of any savings scheme from FNPF or are aware of the benefits of FNPF. AGE is an ordinal variable that represents age groupings of the survey participants, LTN reflects the location of the survey participants, GNDR is a nominal variable set to 1 for male and 2 for female, SIZE representing the market vendor's family size, INC represents weekly personal income of the market vendor, DRTN represents duration as a market vendor, SAVNG represents weekly savings, CURRENT SAVNG represents knowledge of current savings products, AVNE represents knowledge of other avenues for savings, and FRMVND measures whether the market vendor is also a farmer or not, and WYRMV measures whether the market vendor would recommend this line of work. AGE, SIZE, INC, DRTN, and SAVNG are ordinal variables with the remainder being nominal variables. Further descriptions are provided in Table 1:

Given the exploratory nature of our research, we do not make any predictions on the expected signs of the explanatory variables. The variables were drawn from a preliminary literature review, which helped formulate the survey questions [40,43,50,52,53,57,58,62]. Correlation analysis was used to determine whether there are any positive or negative relationships between our variables that could provide us with meaningful relationships. The regression analysis was used to determine factors that determine awareness of savings with FNPF and factors that determine awareness of savings benefits with awareness. They were tested separately to understand whether there are any significant differences between awareness of savings with FNPF and awareness of the benefits of savings with FNPF.

The Robust Least Squares (RLS) technique using the M-estimation method is used for analysis. This is a powerful statistical technique used to estimate regression models while mitigating the influence of outliers and violations of the assumptions underlying ordinary least squares (OLS) regression. Unlike OLS, which minimizes the sum of squared residuals, M-estimation minimizes a function of the residuals that reduces the weight of outliers, providing more reliable parameter estimates in the presence of heteroscedasticity or non-normal error distributions. This method is implemented to specify different weighting functions (such as Huber, Tukey, or Andrews weights), allowing researchers to tailor the robustness of the estimation to their specific data characteristics. By employing the M-estimation methodology, researchers can achieve accurate and robust inferences, enhancing the reliability of analyses.

**Table 1. Variable Definitions.**

| Symbol | Variable | Measurement |
|---|---|---|
| FNPF AWARENESS | Are you aware of any savings scheme from FNPF? | 0 = No Response, 1 = No, 2 = Yes |
| FNPF BENEFIT | Are you aware of the benefits of savings with FNPF? | 0 = No Response, 1 = No, 2 = Yes |
| AGE | Age | 1 = 18–20 Years, 2 = 21–25 Years, 3 = 26–30 Years, 4 = 31–35 Years, 5 = 36–40 Years, 6 = 41–45 Years, 7 = 46–50 Years, 8= 51–55 Years, 9= 56–60 Years, 10 = 61–65 Years, 11 = 66–70, 12 = 71–75 Years |
| LTN | Location | 1 = Sigatoka, 2 = Rakiraki, 3 = Tavua |
| GNDR | Gender | 1 = Male, 2 = Female |
| SIZE | Family Size | 1 = Single, 2 = 2–3 Family Members, 3 = 4–5 Family Members, 4 = 6–7 Family Members, 5 = 8–9 Family Members, 6 = 10–11 Family Members |
| INC | Personal Income | 0 = No Response, 1 = $1-$50, 2 = $51-$100, 3 = $101-$150, 4 = $151-$200, 5 = $201-$250, 6 = $251-$300, 7 = $301–350, 8 = $351-$400, 9 = $401-$450, 10 = $451-$500, 11 = Greater than $501 |
| DRTN | Duration as Market Vendor | 1 = 1 Year, 2 = 2–5 Years, 3 = 6–10 Years, 4 = 11–15 Years, 5 = 16–20 Years, 6 = 21–25 Years, 7 = 26–30 Years, 8 = 31–35 Years, 9 = 36–40 Years |
| SAVNG | How much do you save weekly? | 1 = $1-$20, 2 = $21-$40, 3 = $41-$60, 4 = $61-$80, 5 = $81-$100, 6 = $101-$120, 7 = $121-$140, 8 = $141-$160, 9 = $161-$180, 10 = $181–200, 11 = More than $201 |
| CURRENT SAVNG | Where do you usually save? | 1 = Bank, 2 = Home, 3 = Market Clubs, 4 = Invest in a Business, 5 = Bank and FNPF |
| AVNE | What other avenues of Saving are you aware of? | 1 = FNPF, 2 = Bank, 3 = FNPF, 4 = Yes, 5 = Bank/FNPF, 6 = Market Investment, 7 = Market Clubs, 8 = LICI (Life Insurance) |
| FRMVND | Vendor Only or Farmer and Vendor | 1 = Farmer and Market Vendor, 2 = Market Vendor |
| WYRMV | Would you recommend your children or relatives or anyone to become a market vendor? | 0 = No Response, 1 = No, 2 = Yes, 3 = Unsure |

## Objective 2

Objective 2 is fulfilled by thematic analysis of the interview questions on whether vegetable market vendors prefer tailor-made FNPF programs. The thematic analysis is discussed in the results section. Codes were assigned to the responses to the interview questions. Upon coding all responses, the codes were used to determine underlying themes. The themes were reviewed to check if all codes had been included. The frequency of each of the themes is noted and denotes their importance.

## Results

### Descriptive statistics

Table 2 describes the descriptive statistics of this study. Convenience sampling was used to gather data from 71 respondents composed of: 28 from Sigatoka market, 21 from Rakiraki market and 22 from Tavua market. To observe the measure of central tendency for the data-set, normal distribution is assessed given the possibility of outliers. The values of Skewness, Jarque-Bera and P- Value are considered guided by the rule of thumb If skewness between -0.5 and 0.5 the data is fairly symmetrical, if skewness between -1 and -0.5 or 0.5 and 1 data is moderately skewed and values greater than -1 or 1 signal highly skewed. For notable vari-ables, the distribution ranges from fairly symmetrical to moderately skewed; thus, the mean is discussed.

The average age of the vegetable market vendor is between 45 and 46 years. This confirms that the vendors are in the later stage of work life and require significant consideration on retirement plans and regular sources of income after retirement. The retirement age in Fiji

**Table 2. Descriptive Statistics.**

| VARIABLES | Mean | Median | Maximum | Minimum | Std. Dev. | Skewness | Kurtosis | Jarque-Bera | Probability | Observations |
|---|---|---|---|---|---|---|---|---|---|---|
| Age | 6.58 | 6 | 12 | 2 | 2.18 | 0.19 | 2.54 | 1.06 | 0.59 | 71 |
| Location | 1.92 | 2 | 3 | 1 | 0.84 | 0.16 | 1.45 | 7.39 | 0.02 | 71 |
| Gender | 1.69 | 2 | 2 | 1 | 0.47 | -0.82 | 1.68 | 13.19 | 0.00 | 71 |
| Family Size | 2.92 | 3 | 6 | 2 | 0.91 | 0.86 | 3.58 | 9.79 | 0.01 | 71 |
| Weekly Personal Income | 1.99 | 2 | 11 | 0 | 2.36 | 1.89 | 7.30 | 97.12 | 0.00 | 71 |
| Duration as Market Vendor | 3.32 | 3 | 9 | 0 | 2.10 | 0.51 | 2.42 | 4.11 | 0.13 | 71 |
| Weekly Savings | 2.42 | 2 | 11 | 0 | 2.38 | 2.20 | 7.59 | 119.70 | 0.00 | 71 |
| Where do you save? | 1.25 | 1 | 5 | 0 | 0.75 | 3.02 | 13.07 | 408.40 | 0.00 | 71 |
| Farmer and Market Vendor or Market Vendor Only | 1.30 | 1 | 2 | 0 | 0.72 | -0.51 | 2.05 | 5.73 | 0.06 | 71 |
| Would you recommend Market Vendor to others? | 1.54 | 2 | 3 | 0 | 0.75 | -0.22 | 2.71 | 0.83 | 0.66 | 71 |
| Other Avenues of Savings? | 2.99 | 3 | 8 | 0 | 1.61 | -0.18 | 4.13 | 4.19 | 0.12 | 71 |
| FNPF Savings Awareness | 1.58 | 2 | 2 | 0 | 0.58 | -0.98 | 2.96 | 11.33 | 0.00 | 71 |
| FNPF Savings Benefit Awareness | 1.66 | 2 | 2 | 0 | 0.51 | -1.02 | 2.77 | 12.36 | 0.00 | 71 |

was revised from 55 to 60 years in 2023. Gender balance was always a priority, the average respondents were females, but the researchers observed that most market vendors willing to be respond were females, and this could also be down to the fact that there were more female vegetable market vendors in the three markets surveyed. On average, the family size of the market vendors were between 4 and 5 members, with median income ranging between $51 and $100.

The average duration as a market vendor is between 6 years to 10 years. Therefore, the vendors must accumulate sufficient savings in the span of 6 to 10 years or invest wisely into schemes to hedge risks (disruptions in sale activities) and ensure smooth consumption after retirement. The median savings for the vendors is $21 to $40 with preferred savings avenues such as Banks. While banks offer very low returns, market vendors use this avenue based on ease of depositing and ease of withdrawal, especially in the age of SMS and Internet banking. The market vendors are generally aware of savings with FNPF and the benefits of savings with the FNPF. In addition, there was a strong consensus by the market vendors for tailor-made savings schemes from FNPF. Moreover, a moderate number of respondents had reservations in recommending others to work as market vendors given the challenges (low income, disaster risk, price fluctuations and no retirement schemes).

## Correlation analysis

As stipulated in Table 3 below, there were 16 correlations between the 13 variables used for this study. The correlations were identified using Mukaka's [64] rule of thumb.

As described in Table 2, the first notable relationship indicates a moderate positive relationship between FNPF benefit and FNPF savings benefit awareness. This indicates that those who benefit from the FNPF are those who are aware of the benefits of savings with FNPF. The second notable relationship indicate a low negative correlation between age and FNPF benefit thus, older market vendors have low awareness on the FPNF benefit savings. As older market vendors are less aware of FNPF benefits, the underlying reason for this could be the lack of financial literacy, as those in this line of work are less educated. The third notable relationship indicates low FNPF Savings awareness and low FNPF Savings benefit awareness in Tavua and Rakiraki. Tavua and Rakiraki can be considered the least economically developed of other towns and cities in this study and this can be attributed

**Table 3. Correlation Matrix.**

| Correlation/P-value | FNPF AWARE | FNPF BENEFIT | AGE | LTN | GNDR | SIZE | INC | DRTN | SAVNG | CURRENT SAVNG | FRMVND | WYRMV | AVNE |
|---|---|---|---|---|---|---|---|---|---|---|---|---|---|
| FNPF AWARE | 1 | | | | | | | | | | | | |
| | ----- | | | | | | | | | | | | |
| FNPF BENEFIT | **0.63** | 1.00 | | | | | | | | | | | |
| | 0.00 | ----- | | | | | | | | | | | |
| AGE | -0.13 | **-0.35** | 1.00 | | | | | | | | | | |
| | 0.27 | 0.00 | ----- | | | | | | | | | | |
| LTN | **-0.28** | **-0.51** | 0.08 | 1.00 | | | | | | | | | |
| | 0.02 | 0.00 | 0.50 | ----- | | | | | | | | | |
| GNDR | -0.18 | -0.03 | 0.11 | 0.08 | 1.00 | | | | | | | | |
| | 0.14 | 0.83 | 0.37 | 0.52 | ----- | | | | | | | | |
| SIZE | -0.12 | 0.06 | **-0.21** | **0.23** | 0.14 | 1.00 | | | | | | | |
| | 0.30 | 0.61 | 0.07 | 0.05 | 0.24 | ----- | | | | | | | |
| INC | -0.13 | **-0.31** | -0.07 | **0.45** | -0.13 | 0.04 | 1.00 | | | | | | |
| | 0.28 | 0.01 | 0.58 | 0.00 | 0.27 | 0.74 | ----- | | | | | | |
| DRTN | **0.27** | 0.14 | **0.30** | -0.20 | **-0.23** | **-0.35** | -0.02 | 1.00 | | | | | |
| | 0.02 | 0.23 | 0.01 | 0.09 | 0.05 | 0.00 | 0.87 | ----- | | | | | |
| SAVNG | 0.11 | -0.08 | -0.16 | 0.13 | 0.06 | 0.10 | **0.27** | 0.02 | 1.00 | | | | |
| | 0.36 | 0.50 | 0.18 | 0.30 | 0.65 | 0.43 | 0.02 | 0.85 | ----- | | | | |
| CURRENT SAVNG | 0.18 | 0.08 | 0.01 | 0.08 | -0.10 | -0.05 | -0.09 | 0.19 | 0.01 | 1.00 | | | |
| | 0.12 | 0.52 | 0.91 | 0.51 | 0.41 | 0.67 | 0.47 | 0.11 | 0.93 | ----- | | | |
| FRMVND | 0.06 | 0.04 | 0.10 | **0.23** | -0.02 | **0.26** | 0.23 | -0.05 | 0.07 | 0.18 | 1.00 | | |
| | 0.60 | 0.72 | 0.42 | 0.05 | 0.86 | 0.03 | 0.06 | 0.71 | 0.58 | 0.14 | ----- | | |
| WYRMV | 0.17 | 0.11 | -0.08 | -0.02 | -0.09 | 0.07 | 0.04 | -0.11 | -0.14 | **0.24** | 0.07 | 1.00 | |
| | 0.17 | 0.38 | 0.52 | 0.88 | 0.45 | 0.58 | 0.76 | 0.36 | 0.26 | 0.05 | 0.55 | ----- | |
| AVNE | **0.24** | 0.08 | 0.12 | -0.13 | -0.04 | -0.22 | -0.21 | 0.20 | -0.05 | 0.16 | -0.03 | 0.11 | 1.00 |
| | 0.04 | 0.50 | 0.30 | 0.29 | 0.72 | 0.07 | 0.08 | 0.10 | 0.70 | 0.19 | 0.78 | 0.35 | ----- |

**Notes:** 0.90 to 1.0 very high positive correlation, 0.70 to 0.90 high positive correlation, 0.50 to 0.70 moderate positive correlation and 0.30 to 0.50 low positive correlation.

-0.90 to -1.0 very high negative correlation, -0.70 to -0.90 high negative correlation, -0.50 to -0.70 moderate negative correlation and -0.30 to -0.50 low negative correlation [64].

to lack of financial literacy attributed to lower education levels. Furthermore, the weekly income earned by the vendor is larger in Sigatoka compared to Tavua and Rakiraki. Sigatoka is considered to be more economically developed of the other two towns. Moreover, higher weekly income restricts market vendors from exploring FNPF benefit. This presents the vendors with the challenges of mitigating risk in times of disaster as earned income will vary based on market conditions.

It is likely that older people prefer to take up the role of market vendor as the job activity of market vendors is not very demanding giving older vendors' lot of flexibility (preferred days to sell). On the other hand, market vendors with big family size may look for alternative source of income (quit being a market vendor) as income from vegetable proceeds may not be sufficient. There is low positive relationship between savings and income as vendors attempt to save lower amount of income given the income generated from sale proceeds are not so high. The results also indicate that those market vendors who are aware of savings avenues are also aware of FNPF savings. Being financially literate provides the market vendors the competitive edge in securing a better future.

## Regression analysis

Table 4 presents the results of the FNPF awareness and FNPF benefits model. Among the predictors of the FNPF awareness model, we find statistically significant effects from Age, location, weekly personal income, duration as a market vendor, and whether the market vendor is also a farmer. Age has a negative association on FNPF awareness suggesting that younger market vendors are generally aware of FNPF. Location has a negative association with FNPF awareness. This is not surprising because amongst the western division municipalities considered, Rakiraki and Tavua are generally more less developed than Sigatoka suggesting that the level of local/municipal development may plays a key role in determining whether market vendors are aware of the benefits of FNPF.

Surprisingly, weekly personal income as a negative association with FNPF awareness. Insights on potential reasons are offered by the qualitative interview which suggested that there is a level of distrust on financial institutions in Fiji. There is some weight to this argument given that the market vendors quoted the collapse of the National Bank of Fiji in the 1990's. The market vendors indicated that they would like to steer clear of financial institutions in Fiji.

More surprisingly, duration as a market vendor has a positive association with FNPF awareness which appears to conflict with the effect of AGE on FNPF awareness. A closer inspection of the dataset reveals that the duration a person spends as a market vendor appears to be independent of their age given that there is no formal requirement or official retirement age of market vendors.

The final variable that has a significant effect is whether the market vendor is also a farmer. This has a statistically significant effect suggesting that market vendors who are also farmers are aware of FNPF as a superannuation scheme.

Similar results are reported in the FNPF benefits model. We find that age and location are negatively related to the awareness of FNPF benefits, duration as a market vendor has a negative association, and other avenues of savings have a positive awareness. Weekly savings and personal income have negative associations, whereas having the dual role of farmer and market vendor has a positive association.

Generally, about 62% of respondents are aware of savings with FNPF, while about 68% are aware of the benefits of saving with FNPF. This indicates a good level of awareness among market vendors in terms of savings awareness and the benefits of savings awareness with FNPF. This is contrary to the findings of Gough and Hick [52], Loretto et al. [53] and Heenkenda [62].

As described in Table 5, the models generally satisfy the various diagnostic criteria, such as the residuals being free of autocorrelation based on the Breusch-Godfrey serial correlation LM test, the model having the correct functional form based on Ramsey's RESET test, and the residuals normally distributed based on the Jarque-Bera test. However, as we expected, both models had the statistical presence of heteroscedastic error variances based on the Breusch-Pagan-Godfrey test. This was corrected using Huber-White standard errors in the main estimates.

## Thematic analysis

Objective 2 of the study is addressed using a thematic analysis of the qualitative question of why vegetable market vendors prefer a tailor-made superannuation plan. From the 71 respondents, only 36 provided reasons for why they preferred or did not prefer a tailor-made superannuation plan for vegetable market vendors. From the 36 participants' who responded, we coded the responses and identified four themes. These themes included future planning and

**Table 4. Main Results.**

| VARIABLE | COEFF. | STD. ERROR | Z-STAT. | PROB. |
|---|---|---|---|---|
| **Panel A. FNPF awareness model** | | | | |
| AGE | -0.1257*** | 0.0283 | -4.4390 | <0.01 |
| LTN | -0.1748** | 0.0828 | -2.1116 | 0.03 |
| GNDR | -0.0022 | 0.1057 | -0.0210 | 0.98 |
| SIZE | 0.0936 | 0.0670 | 1.3965 | 0.16 |
| INC | -0.0391* | 0.0211 | -1.8536 | 0.06 |
| DRTN | 0.0940*** | 0.0314 | 2.9974 | <0.01 |
| SAVNG | -0.0262 | 0.0247 | -1.0607 | 0.29 |
| CURRENT SAVNG | -0.0367 | 0.0543 | -0.6756 | 0.49 |
| FRMVND | 0.1337* | 0.0737 | 1.8147 | 0.06 |
| WYRMV | 0.0744 | 0.0652 | 1.1407 | 0.25 |
| AVNE | 0.0465 | 0.0381 | 1.2207 | 0.22 |
| INTERCEPT | 2.0357*** | 0.5103 | 3.9894 | <0.01 |
| **Robust Statistics** | | | | |
| R-squared | 0.3650 | Adjusted R-squared | | 0.2467 |
| Rw-squared | 0.6693 | Adjust Rw-squared | | 0.6693 |
| Akaike info criterion | 139.7991 | Schwarz criterion | | 168.1875 |
| Deviance | 8.2748 | Scale | | 0.2659 |
| Rn-squared statistic | 113.3227 | Prob(Rn-squared stat.) | | <0.01 |
| Non-robust Statistics | | | | |
| Mean dependent var | 1.5774 | S.D. dependent var | | 0.5772 |
| S.E. of regression | 0.6149 | Sum squared resid | | 22.3120 |
| **Panel B. FNPF benefits model** | | | | |
| AGE | -0.1286*** | 0.0259 | -4.9563 | <0.01 |
| LTN | -0.2561*** | 0.0646 | -3.9626 | <0.01 |
| GNDR | 0.0833 | 0.0987 | 0.8443 | 0.39 |
| SIZE | 0.1192** | 0.0470 | 2.5380 | 0.01 |
| INC | -0.0389** | 0.0202 | -1.9280 | 0.05 |
| DRTN | 0.0932*** | 0.0267 | 3.4897 | <0.01 |
| SAVNG | -0.0476** | 0.0215 | -2.2158 | 0.02 |
| CURRENT SAVNG | -0.0200 | 0.0511 | -0.3908 | 0.69 |
| FRMVND | 0.1668*** | 0.0618 | 2.7011 | <0.01 |
| WYRMV | 0.0819 | 0.0568 | 1.4401 | 0.14 |
| AVNE | 0.0321 | 0.0276 | 1.1631 | 0.24 |
| INTERCEPT | 2.0014*** | 0.2946 | 6.7926 | <0.01 |
| Robust Statistics | | | | |
| R-squared | 0.4674 | Adjusted R-squared | | 0.3681 |
| Rw-squared | 0.7233 | Adjust Rw-squared | | 0.7233 |
| Akaike info criterion | 96.7894 | Schwarz criterion | | 129.3643 |
| Deviance | 6.4021 | Scale | | 0.2861 |
| Rn-squared statistic | 111.3261 | Prob(Rn-squared stat.) | | <0.01 |
| Non-robust Statistics | | | | |
| Mean dependent var | 1.6619 | S.D. dependent var | | 0.5055 |
| S.E. of regression | 0.4217 | Sum squared resid | | 10.4930 |

Note: M settings: weight=Bisquare, tuning=4.685, scale=MAD (zero centered); Huber Type III Standard Errors & Covariance. ***,**,* indicate statistical significance at the 1, 5, and 10 percent levels.

**Table 5. Diagnostic Tests.**

| Test | Model 1 Chi-squared Statistic | P Value | Model 2 Chi-squared Statistic | P Value |
|---|---|---|---|---|
| Serial correlation | 3.75 (DF=2) | 0.17 | 0.51 (DF=2) | 0.77 |
| Heteroscedasticity | 27.92 (DF=8)*** | <0.01 | 21.11 (DF=8)*** | <0.01 |
| Functional Form | 0.15 (DF=1) | 0.89 | 2.62 (DF=1) | 0.11 |
| Residual Normality | 2.44 (DF=1) | 0.28 | 3.02 (DF=1) | 0.22 |

Notes: *** indicate statistically significant heteroscedasticity at the 1 percent levels.

retirement, financial security, inconsistent income, risks and challenges and economic and social impact. The most common theme was future planning and retirement. Here, respondents provided comments such as, "to have a secured future" (P-40), "for future when we cannot do market vendoring anymore" (P-45), "for our retirement" (P-53), "for our children to benefit once we retire" (P-54) and "it will assist in paying for school fees for my children and grandchildren when I retire" (P-65). The second most common theme identified was financial security with nine responses aligned to this theme. Respondents stated, "I need a tailor-made plan so I can increase my personal savings" (P-6), "...for ease in savings and getting interest" (P-30), "for safe-keeping" (P-57) and "to benefit from interest gained on savings" (P-58). One other theme was inconsistent income, risks and challenges. This theme consisted mainly of respondents who did not prefer a tailor-made plan. They stated, "I do not have consistent income so I cannot contribute regularly" (P-15), "it can be risky" (P-27), "I will have difficulty in withdrawing if I need my savings" (P-33) and "I do not need a tailor-made plan because cannot save" (P-70). The last theme was economic and social impact. Respondents mentioned that "FNPF should visit us to help us contribute as with current system it is too difficult" (P-34), "we need a tailor-made plan for farmers and market vendors so that we can contribute to the economy" (P-37) and "a tailor-made plan will improve the living conditions of farmers" (P-45). The theme of future planning and retirement indicates that vegetable market vendors are concerned about their future and retirement. However, they have no simple and easy means to make contributions to their superannuation. The second theme of financial security describes the vegetable market vendors' perception of superannuation as a means of financial security in case of emergencies as well as disasters which are common in Fiji [65]. The third theme also highlights some important considerations that the superannuation fund (FNPF) needs to account for when they make a tailor-made plan or for some basic reasons why one is needed. Respondents have highlighted that their inconsistent income is one of the reasons they cannot regularly contribute to superannuation schemes. Additionally, vegetable market vendors cannot regularly withdraw from the fund if needed during disasters and other hardships. These reasons explain why a tailor-made plan is needed for vegetable market vendors and perhaps for agriculture or non-formal sectors. Their inconsistent income should not be a hindrance to contributing, while a mandatory contribution may be required when there is surplus income or savings. The tailor-made plans should also consider that vegetable market vendors and other agriculture and non-formal workers may need to make withdrawals in emergencies, considering the volatile nature of their occupations. This may be the reason why vegetable market vendors prefer to save in a bank and other means over saving in a superannuation fund. Therefore, the tailor-made plan should allow them to make emergency withdrawals in certain circumstances. The final theme of future planning and retirement describes the desire for vegetable market vendors to contribute to superannuation schemes and get annual interest for the future. It will also allow them to contribute to the economy through superannuation fund investments, have emergency funds for emergencies and secure their future upon retirement, just like those in the formal sector.

## Conclusion

Superannuation funds provide a sense of security to members upon retirement and even pre-retirement with benefits such as housing, medical, funeral, and unemployment in the case of FNPF. Therefore, all nationals should have an equal chance to benefit from this scheme. This study attempted to understand vegetable market vendors' perceptions of the FNPF superannuation savings scheme. This study interviewed 71 vegetable market vendors in Sigatoka, Tavua, and Rakiraki concerning their perceptions of FNPF savings schemes.

This study generally found that while 95.8% of these vegetable market vendors indicated that they save, 83.1% of them save in banks only, which may indicate that they do not see saving with FNPF as a benefit or option. 62.3% of the respondents know an option to save with FNPF, while 67.6% indicated that they know the benefits of savings with FNPF. While the previous statistics show above-average awareness, it is strange that 83.1% save in banks. The underlying reason is the lack of motivation and intention to adopt FNPF as a savings option. This can be due to the fact that it is easier for vegetable market vendors to save in a bank than with FNPF. Thus, the FNPF needs to make a tailor-made pension contribution scheme for vegetable market vendors, farmers and those in the informal sector.

Our findings have significant policy implications. Firstly, 94.4% of the respondents indicate that they will welcome a tailor-made savings scheme from FNPF. Considering this and other findings, Fiji and FNPF should tailor-make a pension scheme for vegetable market vendors, farmers and those in the informal sector. The current FNPF structure is a pro-formal sector where employees and employers have a portion of their income and contribution by the employer directly deposited into an employee's FNPF account. This does not suit informal sector workers such as market vendors because their income fluctuates, and there are no other formal avenues in which they can easily and flexibly contribute to their FNPF account like their formal sector counterparts. This tailor-made plan needs to be flexible and easy for informal sector workers to contribute to. One way this can be achieved is through the only formal payment that market vendors make, the market vendor fees that are paid to the municipal council. The respective stakeholders such as FNPF, respective ministries and local municipal councils, can liaise and add an optional or fixed contribution to the market vendor fees market vendors will easily be able to contribute. After collection of the fee, local municipal councils can credit the amount to the FMPF account of respective market vendors. This creates a simple, flexible and easy way for market vendors to make FNPF contributions. This is important because market vendors work Mondays through to Saturdays and are unable to leave their market stalls to deposit FNPF contributions.

The need for a tailor-made pension scheme for farmers has been recommended in prior studies such as Jansuwan and Zander [50]. While awareness is high in market vendors towards FNPF savings and awareness of savings benefits with FNPF, their intention to save with FNPF is low. The underlying reasons for the lack of intention could be fluctuating income, lack of withdrawal availability and a structure that does not allow market vendors to easily contribute to their FNPF accounts. Therefore, the need for a tailor-made FNPF plan is justified as discussed above. This finding reveals that contrary to the lack of and the need for awareness of pension schemes for farmers by Gough and Hick [52], Loretto et al. [53] and Heenkenda [62], the intention to save in superannuation savings take the centre stage when awareness levels have been attained. Both these implications must be transformed into policies as occupational tailor-made pension schemes will prevent workers from quitting [56], improve the mental well-being of the rural elderly [57], better socio-economic and health status [59], and improve land use efficiency by shifting operations of farms from the elderly farmers who are found to be less productive. The entire spectrum for superannuation funds is to provide retirement benefits to its members. This should not be based on one's line of work

or the amount one earns, and therefore a tailor-made savings scheme for vegetable market vendors, farmers, and those involved in informal sectors should be developed.

Considering the debate on the role of financial literacy, despite the vast literature on developed economies, it is comprehensible that financial progress cannot be the absence of financial literacy. Our findings showed that the market vendors know the importance of saving. However, the avenues to save are restricted, with 83.1% saving in banks. Fiji needs to maintain its financial literacy programs while also strategizing means to increase an individual's intention to save, especially in superannuation funds. The Reserve Bank of Fiji has undertaken an active role in setting up the national financial inclusion taskforce together with a financial literacy mascot, 'Vuli the Vonu'. The policy implication is that we need to move away from a 'one shoe fits all financial literacy program' and redefine the financial literacy program to be tailor-made. More importantly, there needs to be a tailor-made savings product that is easy and flexible for informal sector workers such as farmers and market vendors.

Secondly, the policy setup should be redirected from a macro to a micro perspective. That is, instead of policies for the agriculture sector, the policies must focus on managing different industries within the agriculture sector as they have a modus operandi. This would provide a more stable platform to market vendors and deter them from withdrawing from the agriculture sector. Finally, bearing the resource constraint posed on a small nation, the local towns and city councils, through the market master/administrator, can engage in a Memorandum of Understanding with FNPF to facilitate the monthly collection of contributions for FNPF from willing market vendors. These implications can transcend to countries with similar situations, such as Fiji. This ensures that retirement benefits are available to agriculture workers so that the industry retains workers and more people are attracted to boost agricultural output.

Nevertheless, a major limitation of this study was that certain municipalities did not permit us to conduct our interviews. This severely limited our ability to gather the necessary data on market vendors from Ba, Nadi, and Lautoka. In municipalities where we were allowed to interview market vendors, market vendors were also reluctant to participate, so convenience sampling had to be utilized. Due to time and budget constraints, we could not extend our survey to include market vendors in the central and northern divisions. As such, it is possible that the sample is subject to selection bias where the views of the observed market vendors are representative of the views of market vendors in the western division of Fiji. Future research should consider a more representative sample. Nevertheless, the lack of access to superannuation packages is a pervasive issue across all major vegetable markets in Fiji. This brings to light another limitation of this study. Specifically, the sample size is a bit small for regression analysis. A power analysis for regression with eight regressors with a moderate effect size of 0.15 and minimum statistical power of 0.8 returns 108 respondents [66]. Future research could overcome this limitation and explore further dimensions, such as the willingness of the FNPF to accommodate the requests of market vendors, namely having a special FNPF employee collect and update the contributions of the market vendors.

## Supporting information

**S1 File. FHEC FNPF.**
(XLSX)

**S2 Appendix. Survey instrument.**
(DOCX)

## Author contributions

**Conceptualization:** Navneel Shalendra Prasad.

**Data curation:** Nikeel Nishkar Kumar.

**Formal analysis:** Nikeel Nishkar Kumar.

**Funding acquisition:** Navneel Shalendra Prasad.

**Investigation:** Navneel Shalendra Prasad, Amit Prakash.

**Methodology:** Amit Prakash, Nikeel Nishkar Kumar.

**Project administration:** Navneel Shalendra Prasad.

**Resources:** Amit Prakash.

**Software:** Amit Prakash.

**Supervision:** Navneel Shalendra Prasad.

**Validation:** Navneel Shalendra Prasad, Nikeel Nishkar Kumar.

**Visualization:** Navneel Shalendra Prasad, Amit Prakash.

**Writing – original draft:** Navneel Shalendra Prasad.

**Writing – review & editing:** Navneel Shalendra Prasad, Nikeel Nishkar Kumar.

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
