## [Decision Letter · Decision Letter 0]

30 Jan 2024

PONE-D-23-21438

Informal Workers' Perceptions of Retirement Planning in Developing Countries

PLOS ONE

Dear Dr. Prasad,

Thank you for submitting your manuscript to PLOS ONE. After careful consideration, we feel that it has merit but does not fully meet PLOS ONE’s publication criteria as it currently stands. Therefore, we invite you to submit a revised version of the manuscript that addresses the points raised during the review process.

The manuscript has been assessed by two reviewers, and their comments are appended below.

The reviewers have raised a number of concerns that require attention to ensure the manuscript meets our submission guidelines (https://journals.plos.org/plosone/s/submission-guidelines). The introduction should provide background that puts the manuscript into context and allows readers outside the field to understand the purpose and significance of the study. Please also pay close attention to addressing reviewer 2's concerns regarding the methodology of the manuscript. Our publication criteria (https://journals.plos.org/plosone/s/criteria-for-publication) indicate that the methods and reagents must be described in sufficient detail for another researcher to reproduce the experiments described and sample sizes must be large enough to produce robust results.

Please note, at this time we cannot guarantee future acceptance of this manuscript. Continued consideration of the manuscript is contingent upon thorough revision. Could you please revise the manuscript to carefully address the concerns raised?

We look forward to receiving your revised manuscript.

Kind regards,

Laura Kelly

Division Editor

PLOS ONE

Journal Requirements:

3. In the ethics statement in the Methods, you have specified that verbal consent was obtained. Please provide additional details regarding how this consent was documented and witnessed, and state whether this was approved by the IRB

6. Please ensure that you refer to Figure 2 in your text as, if accepted, production will need this reference to link the reader to the figure.

7. Please include your tables as part of your main manuscript and remove the individual files. Please note that supplementary tables (should remain/ be uploaded) as separate "supporting information" files

Reviewers' comments:

Reviewer's Responses to Questions

**Comments to the Author**

1. Is the manuscript technically sound, and do the data support the conclusions?

Reviewer #1: Yes

Reviewer #2: Partly

2. Has the statistical analysis been performed appropriately and rigorously? 

Reviewer #1: Yes

Reviewer #2: No

3. Have the authors made all data underlying the findings in their manuscript fully available?

Reviewer #1: Yes

Reviewer #2: Yes

4. Is the manuscript presented in an intelligible fashion and written in standard English?

Reviewer #1: No

Reviewer #2: Yes

5. Review Comments to the Author

Reviewer #1: The paper should not be published as it is because of inconsistent use of capitalization make the paper unintelligible. Financial planning and other terms are sometimes capitalized and sometimes not capitalized. These mistakes are critical mistakes in grammar. Even more frustrating is that the authors have a unique opportunity to describe the Fiji pension system within the context of the Fji labor market. However the authors assume readers know the basic facts about Fiji. The section on the superannuation scheme is completely impenetrable. The authors should imagine sitting in a room of intelligent pension experts and seek to describe the system. After reading this paper carefully I absolutely have no idea what the Fiji pension system is like. The authors also states they have established causality between financial literacy and subjective willingness to contribute to the Fiji system. But this is not made clear. We always need papers on the world‘s pension systems and the inclusion of informal workers into social insurance schemes is probably the most important issue facing modern market economies. But this paper fails at describing, even the simplest terms, with the system is in Fiji. So frustrating. I would love to know about the Fiji superannuation schemes and policies that would bring in informal sector workers into the schemes.

Discussion on farming and the relationship between that and market vendors is not clear to me. And the authors should spend more time describing the actual Fiji superannuation system and what the market offenders are supposed to understand, instead of writing about United Nation’s millennium goals and general food shortages.

Reviewer #2: This paper examines whether and to what extent informal sector workers (i.e., vegetable market venders, or farmers) in Fiji engage in retirement savings—primarily by evaluating their awareness and access to Fiji National Provident Fund (FNPF). In doing so, the authors deploy several empirical estimation strategies including the OLS regression, correlation analysis, and principal component analysis on a first-hand survey (case studies?) of 71 respondents. The authors find that most respondents regarded FNPF in a positive light, with 60-70% of the respondents being aware of the FNPF scheme and its benefits. Yet, 94% of them still preferred an alternative pension scheme that is better tailored for them. The authors also find that age, gender, years of experience (duration) in the current job, and availability of other venues for savings strongly predicted the likelihood of being aware of FNPF schemes.

While I see the value in the authors’ introducing a primary dataset on an underexplored group of informal agricultural sector workers in Fiji, I find several limitations in the overall framing (i.e., motivation for examining FNPF) of the research questions, estimation strategies, and interpretation of the results. Moreover, the authors equate ‘awareness’ of a government-sponsored pension/superannuation scheme (i.e., FNPF) with ‘financial literacy’—when these two are discussed as two distinct concepts in the current literature. Detailed comments are provided as an attachment.

6. PLOS authors have the option to publish the peer review history of their article (what does this mean? ). If published, this will include your full peer review and any attached files.

**Do you want your identity to be public for this peer review?** For information about this choice, including consent withdrawal, please see our Privacy Policy .

Reviewer #1: No

Reviewer #2: No

---

## [Author Response · Author response to Decision Letter 1]

23 Apr 2024

Reviewer response document is attached in the revision submission.

---

## [Decision Letter · Decision Letter 1]

7 Jun 2024

PONE-D-23-21438R1Informal Workers' Perceptions of Retirement Planning in Developing CountriesPLOS ONE

Dear Dr. Prasad,

Thank you for submitting your manuscript to PLOS ONE. After careful consideration, we feel that it has merit but does not fully meet PLOS ONE’s publication criteria as it currently stands. Therefore, we invite you to submit a revised version of the manuscript that addresses the points raised during the review process.

**ACADEMIC EDITOR: ** Thank you for submitting your revised manuscript to PLOS ONE. While the manuscript has clearly improved from the original submission, I feel that some of the changes made by the authors still fall short of addressing concerns raised by Reviewer 3 (comments are attached in a separate document). Therefore, we invite you to submit a revised version of the manuscript that addresses Reviewer 3's concerns.

• A marked-up copy of your manuscript that highlights changes made to the original version. You should upload this as a separate file labeled 'Revised Manuscript with Track Changes'. Instead of the document with track changes (difficult to read), please highlight the changed/newly-added sections in red.

We look forward to receiving your revised manuscript. Should you choose to proceed with the revision, please submit your revised manuscript by Jul 18 2024 11:59PM. If you will need more time than this to complete your revisions, please reply to this message or contact the journal office at plosone@plos.org . Kind Regards,

Zeewan Lee

Guest Academic Editor

PLOS ONE

Journal Requirements:

https://ddec1-0-en-ctp.trendmicro.com:443/wis/clicktime/v1/query?url=https%3a%2f%2fjournals.plos.org%2fplosone%2fs%2ffile%3fid%3dwjVg%2fPLOSOne%5fformatting%5fsample%5fmain%5fbody.pdf&umid=d595e888-7493-410c-a420-0c83e1f209b7&auth=8d3ccd473d52f326e51c0f75cb32c9541898e5d5-f7be46a4b3c543781ac30e939380323191ae18e3 and

https://ddec1-0-en-ctp.trendmicro.com:443/wis/clicktime/v1/query?url=https%3a%2f%2fjournals.plos.org%2fplosone%2fs%2ffile%3fid%3dba62%2fPLOSOne%5fformatting%5fsample%5ftitle%5fauthors%5faffiliations.pdf&umid=d595e888-7493-410c-a420-0c83e1f209b7&auth=8d3ccd473d52f326e51c0f75cb32c9541898e5d5-87faa588aaa1d911ceaada18a13f693781898e26

2. If you have not done so, and if necessary, please submit Funding Statement where you declare all the funding or sources of support (whether external or internal to your organization) received during this study, as detailed online in our guide for authors at https://ddec1-0-en-ctp.trendmicro.com:443/wis/clicktime/v1/query?url=http%3a%2f%2fjournals.plos.org%2fplosone%2fs%2fsubmit%2dnow&umid=d595e888-7493-410c-a420-0c83e1f209b7&auth=8d3ccd473d52f326e51c0f75cb32c9541898e5d5-818317a14a2838fce05fa09144a6ff017a57711d.

3. We strongly recommend all authors decide on a data sharing plan before acceptance, as the process can be lengthy and hold up publication timelines. Please note that, though access restrictions are acceptable now, your entire data will need to be made freely accessible if your manuscript is accepted for publication. This policy applies to all data except where public deposition would breach compliance with the protocol approved by your research ethics board. If you are unable to adhere to our open data policy, please kindly revise your statement to explain your reasoning and we will seek the editor's input on an exemption. Please be assured that, once you have provided your new statement, the assessment of your exemption will not hold up the peer review process.

Reviewers' comments:

Reviewer's Responses to Questions

**Comments to the Author**

1. If the authors have adequately addressed your comments raised in a previous round of review and you feel that this manuscript is now acceptable for publication, you may indicate that here to bypass the “Comments to the Author” section, enter your conflict of interest statement in the “Confidential to Editor” section, and submit your "Accept" recommendation.

Reviewer #1: All comments have been addressed

Reviewer #3: (No Response)

2. Is the manuscript technically sound, and do the data support the conclusions?

Reviewer #1: Yes

Reviewer #3: No

3. Has the statistical analysis been performed appropriately and rigorously? 

Reviewer #1: Yes

Reviewer #3: No

4. Have the authors made all data underlying the findings in their manuscript fully available?

Reviewer #1: Yes

Reviewer #3: Yes

5. Is the manuscript presented in an intelligible fashion and written in standard English?

Reviewer #1: Yes

Reviewer #3: No

6. Review Comments to the Author

Reviewer #1: Thank you for your comments and improvements. This is a very important topic. Fijian street vendors attitudes towards pensions. They want them

Reviewer #3: (No Response)

7. PLOS authors have the option to publish the peer review history of their article (what does this mean? ). If published, this will include your full peer review and any attached files.

**Do you want your identity to be public for this peer review?** For information about this choice, including consent withdrawal, please see our Privacy Policy .

Reviewer #1: No

Reviewer #3: No

---

## [Author Response · Author response to Decision Letter 2]

19 Sep 2024

Response document is attached in this submission.

---

## [Editor Report · Decision Letter 2]

2 Jan 2025

PONE-D-23-21438R2Informal Workers' Perceptions of Retirement Planning in Developing CountriesPLOS ONE

Dear Dr. Prasad,

Thank you for submitting your manuscript to PLOS ONE. After careful consideration, we feel that it has merit but does not fully meet PLOS ONE’s publication criteria as it currently stands. Therefore, we invite you to submit a revised version of the manuscript that addresses the points raised during the review process.

We look forward to receiving your revised manuscript.

Kind regards,

Zeewan Lee

Guest Editor

PLOS ONE

Journal Requirements:

Additional Editor Comments :

I would like to thank the authors for their hard work in revising the manuscript. As an action editor (guest editor) and a former reviewer in the first round, I have taken a thorough look at the past reviewers’ comments as well as authors’ responses to the comments. Upon reviewing, I have decided to accept the paper with a minor revision. Please see my comments below:

**1. Better highlight contributions:** In the introduction section, it would be great if the authors could explicitly state what the contributions of this paper is. What is new in this paper that the prior literature has not done? For instance, in line 123-124, the authors stated that they gained first-hand info on the market vendors’ view toward superannuation. Is this a first paper to do so in Fiji?

**2. Literature Review section:** Lines 134-142 are awkward. The authors say this: “It begins by reviewing literature on the association between the informal sector and financial literacy. It then reviews the literature on retirement planning, and superannuation schemes in developing countries. The link between these groupings is that informal sector workers may lack financial literacy.”

2-1. If this is the case, why not combine the three sections to center the discussion on the role of financial literacy? For instance, why not make a section named “informal sector, superannuation, and financial literacy”? The last two sections (“retirement planning” AND “superannuation in dev. countries”) could definitely be combined.

2-2. Try to weave in lines 137-142 inside the actual sections (i.e., informal sector and financial literacy, superannuation, or a combined section).

2-3. How does the current paper fit in the line of literature reviewed in the Lit Review section? How should the readers situate this paper, vis-à-vis all the papers mentioned in the Lit Review section? I would like to see some comments on this, before the authors move onto the Data section.

**3. Data and Methods Section – Objective 1: ** Equation (1) and Equation (2)—lines 343-346—seem to be exactly the same except the outcomes. In such a case, please show only one equation with an outcome listed as ‘outcome’, and explain that the authors test two outcomes in separate regressions.

3-1. Please describe the variables used in complete sentences instead of a table (Table 1). It is more typical to describe them within the text—in a full paragraph.

**4. Data and Methods Section – Objective 2: ** Please elaborate more on the ‘thematic analysis’ of the interview questions. What are the themes or the method of the analysis?

**5. Results – Regression results (Table 4 & Table 5):** The authors need not show too many decimal points, or the z-statistics. It would be clearer if the authors could present coefficients, standard errors, and statistical significance in the form of asterisks next to the coefficients (For instance * if p<0.1, ** if p<0.05, *** if p<0.001).

5-1. Please streamline the tables. The authors don’t need two separate tables just to show two outcomes. These could be combined into one table. This way, readers will be able to compare the results (of the two outcome variables) more easily.

**Minor**

**6. Line 46 – improve citation:** It is great to see more information on the size of the informal sector in Fiji. Please list the year of the cited stats, and if it is from too far back, please acknowledge in a footnote that more recent statistic is not available.

**7. Sample – acknowledge limitations:** Please acknowledge in the Discussion Section that convenience sampling is used. What are some implications of doing so? For instance, would there be any potential selection bias in the sample that the authors could not address in the current study? It would be good to clearly acknowledge it as one of the limitations.
---

## [Author Response · Author response to Decision Letter 3]

20 Feb 2025

Response to Reviewers is attached in this submission.

---

## [Editor Report · Decision Letter 3]

3 Mar 2025

Informal Workers' Perceptions of Retirement Planning in Developing Countries

PONE-D-23-21438R3

Dear Dr. Prasad,

We’re pleased to inform you that your manuscript has been judged scientifically suitable for publication and will be formally accepted for publication once it meets all outstanding technical requirements.

Kind regards,

Zeewan Lee

Guest Editor

PLOS ONE

Additional Editor Comments (optional):

All comments have been sufficiently addressed. Thank you for the revision efforts.
---

## [Editor Report · Acceptance letter]

PONE-D-23-21438R3

PLOS ONE

Dear Dr. Prasad,

I'm pleased to inform you that your manuscript has been deemed suitable for publication in PLOS ONE. Congratulations! Your manuscript is now being handed over to our production team.

Kind regards,

on behalf of

Dr. Zeewan Lee

Guest Editor

PLOS ONE